# Third-party evaluators perceive AI as more compassionate than expert humans
Dariya Ovsyannikova [1], Victoria Oldemburgo de Mello [1,3] & Michael Inzlicht [1,2,3] ✉

Empathy connects us but strains under demanding settings. This study explored how third parties evaluated AI-generated empathetic responses versus human responses in terms of compassion, responsiveness, and overall preference across four preregistered experiments. Participants (N = 556) read empathy prompts describing valenced personal experiences and compared the AI responses to select non-expert or expert humans. Results revealed that AI responses were preferred and rated as more compassionate compared to select human responders (Study 1). This pattern of results remained when author identity was made transparent (Study 2), when AI was compared to expert crisis responders (Study 3), and when author identity was disclosed to all participants (Study 4). Third parties perceived AI as being more responsive—conveying understanding, validation, and care—which partially explained AI's higher compassion ratings in Study 4. These findings suggest that AI has robust utility in contexts requiring empathetic interaction, with the potential to address the increasing need for empathy in supportive communication contexts.

Empathy is crucial for fostering societal unity and effective communication. It allows individuals to balance their own interests with the wellbeing of others through shared experiences and emotions. It can promote cooperation, altruism, and helping behaviors, thereby strengthening social bonds[1–3]. Psychologically, empathy also has a nourishing effect on its recipients, such that people feel validated, understood, and connected when others empathize with them[4,5]. Despite the positive impact of empathy on its recipients, the effort required to express empathy can be costly and burdensome to the empathizer[6,7], making them less likely to respond empathically, a phenomenon known as empathy avoidance and compassion fatigue[6,8,9]. This seems to be particularly apparent in clinical settings, where healthcare professionals may sacrifice some of their ability to empathize in order to avoid burnout[4,8], to manage personal distress[10,11], or to balance their emotional engagement with the need to effectively allocate resources to each client, particularly individuals with complex cases[12].

One consequence of these challenges is that empathy is in short supply, especially as the mental health sector struggles with accessible service and workforce shortages[13] amid the increasing incidence of mental health disorders[14]. Such shortages make the maintenance of compassionate care even more difficult for the currently employed mental health professionals, for whom it serves as one of several key responsibilities[8]. While empathy is often understood as a dynamic process that originates from the experience of the empathizer[1,3,4], less is known about its effects on the perceivers of empathic support. Considering this and the challenges of meeting societal needs for empathy, here, we compare the quality of written empathic

responses generated by Artificial Intelligence (AI) to select and expert humans. We ask if AI can match or even exceed the quality of responses made by human empathizers and examine the conditions under which people are more likely to prefer an empathetic response from an AI over a human.

In response to the gap between the supply and demand of empathy, scientists have asked if AI could provide consistent and quality supportive care. Despite arguments that AI cannot experience empathy or feel emotions[15], it can *express* empathy by generating responses or behaviors that appear to reflect empathic concern[16] or the intention to alleviate distress[8]. As such, scientists have begun exploring the effectiveness of AI powered by large language models in providing empathic support[16–20]. These investigations, through methods ranging from third-party evaluations[17] to direct recipient feedback[19,20], reveal that AI can be rated as comparable to, if not superior in, expressing empathetic support. For example, in a recent study[17], researchers compared the perceived quality and level of empathy in ChatGPT's responses to public questions generated on a Reddit forum (r/AskDocs) to responses made by verified human physicians through third party ratings made by healthcare professionals. It was found that chatbot responses were rated significantly more empathic and of higher quality than physician responses[17]. Interestingly, chatbot responses were also significantly longer than physician responses, perhaps reflecting the difficulties for humans to convey empathy through written text, particularly when these responses are made by healthcare professionals, who may experience competing demands and time constraints[12,21–23].

[1]Department of Psychology, University of Toronto, Toronto, ON, Canada. [2]Rotman School of Management, University of Toronto, Toronto, ON, Canada. [3]These authors contributed equally: Victoria Oldemburgo de Mello, Michael Inzlicht. ✉e-mail: michael.inzlicht@utoronto.ca

Several lines of evidence illustrate the potential benefits of interacting with an empathic AI. The fact that AI interactions are anonymous and that they involve machines and not humans can facilitate greater disclosure of personal information[24,25], perhaps because chatbots are not inherently judgmental and thus do not evoke a fear of feeling criticized[19]. The latter effect is particularly important, as both the act and degree of self-disclosure have been experimentally demonstrated to increase and deepen subsequent disclosures, increase perceived intimacy and enjoyment of the interaction, as well as increase feelings of trust[24]. Together, these elements and associated outcomes of AI interactions might explain why interacting with artificial agents might provide some social benefits to people[18].

Yet, the receptiveness to AI-generated empathic responses could be influenced by the recipient's awareness of and preconceived attitudes towards receiving support from non-human entities. For instance, people's perception of empathy expressions from AI could be rooted in an awareness that AI, unlike humans, lacks genuine emotional experience[15] and thus cannot actually care; being unmoved by empathic AI statements might then reflect warranted skepticism about its capabilities regardless of its actual effectiveness[20]. Simultaneously, general attitudes towards AI, related to factors such as personality, conspiracy mindset[26], and religiosity[27], among others, may play a critical role in the evaluation and acceptance of AI.

One recent study investigated differences in people's perceptions of feeling heard after receiving human or AI-generated responses that were or were not transparently labeled[20]. AI responses were generally evaluated as more emotionally supportive and responsive than human responses. However, the AI advantage disappeared when participants believed that they were responded to by AI, such that their ratings of feeling heard and understood were higher when they believed that the responses came from a human. Critically, when AI and human responses were accurately labeled, participants reported equivalent perceptions of feeling heard and understood by either agent[20], suggesting that the benefits reaped from empathic AI interactions can occur even after accounting for the drop in perceived response quality, when people are made aware that they are not interacting with another human. Further research found that the mere act of emotional disclosure over a 25-minute conversation carried numerous emotional, psychological, and relational benefits, irrespective of whether participants believed they were conversing with an AI or human agent[19]. Collectively, while these findings highlight the nuances of human reactions to AI-generated content, they challenge the notion that human interaction is irreplaceable in empathic exchanges and further suggest that attitudes towards empathy-expressing AI can improve with increasing familiarity and time.

Despite preliminary evidence that AI responses are rated as being greater or equal in empathy to human responses, there are a few limitations to this initial work. First, given the ethical requirements of consent and transparency in the use of AI[16,28], studies need to compare responses from humans *versus* AI, both when participants are blind to the source and fully aware of it. Doing so allows for the generalizability of empathic AI preference to ethical and legal contexts and allows for the investigation of the effects of AI aversion[29]. Second, the present literature is limited in using laypersons to generate empathic responses that are then compared with AI responses[20]. In other words, these participants do not receive formal training in providing empathic support and/or do not assume professional roles in providing empathic care. At present, there are no known studies that compare empathic AI to trained "experts" of empathy or even samples selected for being particularly empathic, especially individuals working in the mental healthcare sector.

Here, in a series of four preregistered studies, we investigate whether there are significant differences in the way that third-party persons evaluate empathic responses by AI or human agents. We ask: do third-party evaluators rate responses made by AI as more compassionate than responses made by fellow humans selected for being good empathizers? (Study 1); will these differential evaluations hold when the identities of the two sources are made transparent? (Study 2); will they hold when empathic AI is compared to real-life experts of empathic support? (Study 3); and why is transparently labeled AI so good at generating empathic statements? (Study 4). To address the final question, we examine the mediating role of perceived responsiveness in driving judgments of compassion.

We hypothesized that participants would rate the responses generated by AI as more compassionate than those of select and expert human responders. We further hypothesized that participants would rate responses generated by AI as significantly better quality and prefer AI responses to responses generated by select and expert humans in a binary forced-choice scenario. Finally, with respect to Study 4, we further hypothesized that participants would rate the AI-generated responses as more responsive than human-generated responses in terms of communicating care, understanding, and validation[30].

## Methods

Our goal was to assess which agent was better at generating empathic statements: humans or AI. To evaluate this, we compared human-generated or GPT-generated written responses to empathic prompts across four experiments. We first created 10 empathy prompts (first-person descriptions of both positive (5) and negative (5) experiences). In studies 1–3, participants read all 10 empathy prompts describing personal experiences. In study 4, only 6 of the 10 prompts were presented to participants. For each empathy prompt, participants read a pair of potential empathic responses: one human-generated response and one GPT-generated response. Examples of vignettes and responses can be seen in Fig. 1.

To generate the select human responses used in studies 1 and 2, we first piloted a study on our university participant pool and then formally ran the study on Prolific Academic[31]. Ten participants were instructed to read the 10 empathy prompts and generate a compassionate written response to the author of the prompt. Out of the 100 overall responses generated (10 per participant), we asked 3 graduate students and 4 research assistants to rank order the top 5 responders based on how overall compassionate their responses were in terms of quality, emotional salience, relatability, and level of detail. The 5 responders who were ranked in the top 5 most often had their responses selected for use in the studies. Thus, we consider this a select group of empathic responders, as they were first screened and selected based on their overall empathic quality.

In studies 3 and 4, the human response stimuli were obtained from a sample of hotline crisis responders—volunteers trained to respond to psychological crises through telephone calls—whom we considered expert empathizers. These participants were recruited via emails that were internally distributed to all responders within the Distress Centres of Greater Toronto, a multi-helpline organization that offers emotional support to Canadian callers across general and national helplines. Five responders provided written empathic responses to the same 10 empathy prompts as the Prolific Academic sample. All responses were inspected for quality and used in the study, such that each participant only saw one randomly selected option of these 5 responses per vignette.

The AI-generated responses across all studies were created by prompting ChatGPT (model *gpt-4-0125-preview*) with the 10 vignettes describing the emotional experience (one at a time) and asking it to generate an appropriate empathetic response. Given the stochastic nature of ChatGPT, we generated 5 separate responses per vignette. All responses generated by ChatGPT were used in the study. These responses were randomized in the study, such that participants only saw one of these 5 responses per vignette. For a detailed report of how the empathy prompts and responses were generated, see section 1 in the Supplementary Information file.

After reading the empathy prompt and each pair of responses, participants first reported the level of compassion in each response and then selected the one they considered the best at addressing the prompt (response preference). To measure perceived compassion, participants were asked how much each response: a) reflected the emotional state in the prompt, b) was compassionate, and c) was impersonal (reverse-coded). All responses were recorded on a 5-point Likert scale ranging from 1 (Strongly Disagree)

## Negative prompt example

"I am having difficulties with my family relationships. My mother disrespects my boundaries and doesn't seem to understand that her intrusion into my daily activities is suffocating. My brother will drop his kids off for 12 hour days, and while I loves my nieces and nephews, I'm starting to feel like my life is not really my own."

### Human Response

I am sorry that your family has been making you feel this way. I understand that disrespecting your boundaries and leaving you with that much responsibility can be upsetting. You deserve to be treated with more respect and consideration.

### AI Response

It sounds like you're in a really tough spot, feeling overwhelmed by the demands placed on you and struggling with boundaries within your family.[...] Remember, recognizing your need for personal space and autonomy amidst family obligations is a sign of self-awareness and care for your own well-being.

## Positive prompt example

"Today was my final day of therapy! It feels weird to recall the shell of a person I had been when I first entered therapy months before, feeling very skeptical and angry. Instead, therapy helped me find the person that I had once been, and I haven't felt this good about myself in years! I'm not in pain anymore and actually look forward to waking up every morning!"

### Human Response

It is so amazing to hear that you have benefitted from therapy so much! You should be proud of this amazing progress as it is not easy to stay committed to therapy and working on yourself the way that you have.

### AI Response

I want to express how wonderful it is to hear about the positive journey you've had in therapy. Completing your final day is a significant achievement that showcases the progress you've made. [...] Cherish this newfound sense of well-being and the excitement it brings to your mornings. Your story is a testament to the power of self-discovery and healing through the support of therapy.

**Fig. 1 | Example responses to negative and positive prompts from human and AI sources.** Human and ChatGPT-4 generated responses to negative and positive prompts. Each response demonstrates the differing emphases placed on emotional support by humans and AI.

to 5 (Strongly Agree) and averaged per empathic response. Response preference was measured by asking participants which one of the two responses was better at addressing the personal experience in the empathy prompt through a binary forced-choice question, where human responses were coded as 0 and AI responses were coded as 1. We aggregated the ratings for all 10 (or 6) vignette responses to create average scores for compassion and preference for both AI responses and human responses.

In study 4, we measured participants' perceived level of responsiveness for all responses. Participants evaluated the responsiveness of human and AI responses using a 5-point Likert scale, based on facets of *understanding* (paraphrasing the reported experience, further inquiring about the experience, expressing understanding), *validation* (agreeing with the individual, validating their feelings/emotions, using exclamations and judgments), and *caring* (expressing empathy or emotions, offering support, concern, or comfort, and emphasizing the outcome sharing of the individual's scenario and/or circumstances)[30]. Each facet was measured with three items, and responses were recorded using a 5-point Likert scale ranging from 1 (Strongly Disagree) to 5 (Strongly Agree). Responsiveness scores were averaged across these facets. Detailed item measures and figures can be found in sections 1 and 2 of the Supplementary Information file, respectively.

In experiment 1, all participants were left blind to whether each response was generated by a human or AI. In experiments 2 and 3, participants were randomly assigned to either transparent or blind conditions in a between-subject design; in the transparent condition, they were told whether each response was generated by AI or humans; in the blind condition, participants did not see the label for each response, so they could not immediately know which response was generated by a human or AI. Experiment 4 only had the transparent condition, so participants could see the author labels for both empathy sources.

In addition to the response ratings, we also measured (but did not fully analyze) participants' trait level empathy using the Interpersonal Reactivity Index[32]. This was done to explore whether participants' compassion ratings of AI and human-generated responses were moderated by their reported level of trait empathy. More details on this measure can be found in the Supplementary Methods section of the Supplementary Information file, under the *Measures* subsection. Participants were paid £5.25, £4.50, £4.50, and £3.75 (GBP) for their participation in studies 1-4, respectively. All aspects of the present study were approved and undertaken in compliance with the ethical regulations surrounding human research participants set by the Research Ethics Board at the University of Toronto. Informed consent was obtained from all participants, who were all debriefed and compensated following study completion.

### Sampling strategy

In studies 1 and 4, which had a completely within-subject design, we aimed for a sample of 54 participants, given that a power analysis suggests we'd achieve at least 80% power to detect the average effect size in social psychology of d = 0.4[33]. For studies 2 and 3, where we had a mixed design with one between-subject and one within-subject variable, we aimed to run 400 participants, giving us 80% power to detect an interaction as small as f = 0.15 even after dropping the expected number of inattentive participants. A sample of English-speaking participants was recruited on Prolific Academic[31]. Study data was collected between September 2023 and May 2024. After excluding participants that failed one or both attention checks, the distinct sample size for each study consisted of n = 54 and n = 58 participants in studies 1 and 4, respectively, and n = 99 (vs. 98 blind) and n = 121 (vs. 126 blind) participants per condition in studies 2 and 3, respectively. These individuals had an average age of 42.0 years (SD = 13.7) in study 1, 36.2 years (SD = 13.4) in study 2, 37.2 years (SD = 13.6) in study 3, and 37.0 years (SD = 12.3) in study 4. The demographic distribution of our samples in terms of age, gender, and race, with responses to all variables

**Table 1 | Demographic Distributions of Participants in Studies 1–4**

| | Study 1 (*N* = 54) | | Study 2 (*N* = 197) | | Study 3 (*N* = 247) | | Study 4 (*N* = 58) | |
|---|---|---|---|---|---|---|---|---|
| Age in years | | | | | | | | |
| Mean ± SD | 42.0 ± 13.7 | | 36.2 ± 13.4 | | 37.2 ± 13.6 | | 37.0 ± 12.3 | |
| Median | 40 | | 32 | | 33 | | 35 | |
| Minimum - Maximum | 21–76 | | 18–77 | | 18–104 | | 19–64 | |
| Sex | Count | % | Count | % | Count | % | Count | % |
| Female | 25 | 46.3% | 87 | 44.2% | 141 | 57.1% | 37 | 64.9% |
| Male | 28 | 51.9% | 107 | 54.3% | 105 | 42.5% | 20 | 35.1% |
| Prefer not to say | 0 | 0.0% | 1 | 0.5% | 1 | 0.4% | 0 | 0.0% |
| N/A | 1 | 1.9% | 2 | 1.0% | 0 | 0.0% | 0 | 0.0% |
| Race | | | | | | | | |
| Asian | 6 | 11.1% | 32 | 16.2% | 45 | 18.2% | 9 | 15.8% |
| Black | 13 | 24.1% | 27 | 13.7% | 23 | 9.3% | 4 | 7.0% |
| Mixed | 3 | 5.6% | 15 | 7.6% | 10 | 4.0% | 6 | 10.5% |
| White | 31 | 57.4% | 101 | 51.3% | 158 | 64.0% | 35 | 61.4% |
| Other | 0 | 0.0% | 15 | 7.6% | 9 | 3.6% | 3 | 5.3% |
| N/A | 1 | 1.9% | 7 | 3.6% | 2 | 0.8% | 0 | 0.0% |
| Country of Residence | | | | | | | | |
| Canada | 24 | 44.4% | 67 | 34.0% | 187 | 75.7% | 33 | 57.9% |
| United States | 30 | 55.6% | 130 | 66.0% | 60 | 24.3% | 24 | 42.1% |

This table displays the demographic distributions for participants across four distinct studies, detailing age, sex, race, and country of residence. Each study's demographic profile is presented with mean and standard deviation for age, along with the count and percentage breakdown for sex, race, and country of residence. Sample sizes are specified for each study.

provided by participants alongside study data on Prolific Academic[31], is reported in Table 1.

### Statistical analyses

In studies 1 and 4, we ran dependent samples t-tests to evaluate whether the GPT-generated or the human-generated responses were more compassionate. To evaluate response preference, we ran one-sample t-tests where the mean preference (ranging from 0 for human to 1 for AI) was compared against chance (0.5).

In studies 2 and 3, due to the mixed method design, we ran mixed models (using the *afex* package in *R*) to determine the interaction between compassion judgments for GPT *versus* human-generated responses and blind *versus* transparent conditions. For the mediation model in study 4, we used within-person mediation with the *lme4* package, after which we bootstrapped using 1000 samples.

To further explore the results, we divided our vignettes into positive-valenced or negative-valenced—vignettes reporting positive or negative experiences. We used interaction mixed regression models to evaluate whether the vignette valence moderated the relationship between response author (human or AI) and perceived compassion or preference. All analyses were performed on R 4.0.3[34]. Additional information regarding exploratory analyses is reported in Supplementary Notes 1 and 2 of the Supplementary Information File.

### Preregistration

We preregistered studies 1–4 at aspredicted.org. The links for all the studies (preregistration dates included) are provided as follows: study 1 (https://aspredicted.org/ha2av.pdf), study 2 (https://aspredicted.org/c3y4s.pdf), study 3 (https://aspredicted.org/v62tg.pdf), and study 4 (https://aspredicted.org/q5hq9.pdf). All preregistration documents are available at the repository https://osf.io/wjx48/. We originally planned to conduct repeated measures ANOVA in Studies 2, 3, and 4. However, we deviated from this approach and used multilevel models instead, as our data violated Mauchly's test of sphericity, a key assumption of ANOVA. As a robustness check, we also ran ANOVAs, which are reported in Supplementary Note 3

of the Supplementary Information File. The ANOVA results were consistent in direction and significance with those from the multilevel models. Additionally, we mistakenly described response preference as a continuous variable, when it was actually a binary forced-choice variable. Given this, the most appropriate analysis was a one-sample t-test against 0.5, rather than the dependent samples t-test originally preregistered. Finally, while we preregistered exploring whether participants' reported trait empathy moderated their compassion ratings for AI or human-authored responses, we found no credible evidence of trait empathy affecting compassion ratings for either response source. We nevertheless report this finding for study 1 in Supplementary Note 3 of the Supplementary Information File. Across all studies, data distribution was assumed to be normal, but this was not formally tested. However, given our use of multi-level models, which are robust to non-normal data, this assumption was not critical.

### Reporting summary

Further information on research design is available in the Nature Portfolio Reporting Summary linked to this article.

## Results

We initially hypothesized that participants would rate the AI-generated responses as more compassionate than the human-generated responses. We also hypothesized that the AI responses would be preferred over the human responses. The two preregistered hypotheses were confirmed across all four experiments. The results for the compassion hypothesis are summarized for all four studies in Fig. 2. The findings for response preferences across studies 1–4 are summarized in Fig. 3.

### Experiment 1

The AI-generated responses (M = 4.08, SD = 0.59) were rated as significantly more compassionate than the select human-generated responses (M = 3.50, SD = 0.524), t(53) = 5.36, *p* < 0.001, d = 0.73, 95% CI = [0.43, 1.03]. Participants also preferred the AI response over the select human response, t(53) = 5.03, *p* < 0.001, d = 0.68, 95% CI = [0.38, 0.98].

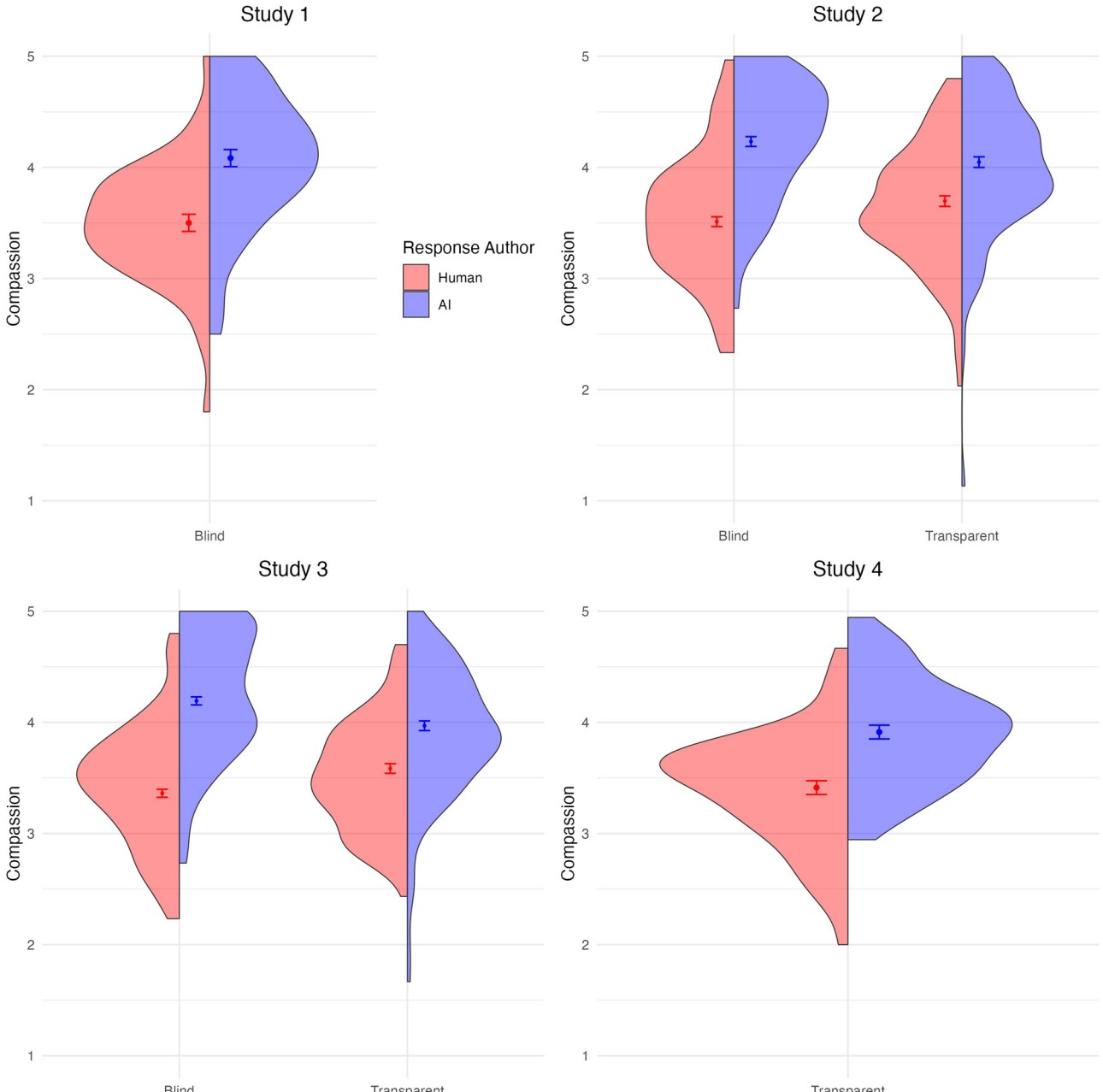

**Fig. 2 | Comparison of compassion ratings by response author across studies 1–4.** Distributions of compassion ratings for responses authored by humans (red) and AI (blue) in four separate studies. Panels represent Studies 1 through 4, with each study split into conditions where the response author label was concealed or transparent to evaluators. Error bars represent the standard error of the mean. The sample sizes are $n = 54$ for Study 1, $n = 197$ for Study 2 ($n = 98$ blind, $n = 99$ transparent), $n = 247$ for Study 3 ($n = 126$ blind, $n = 121$ transparent), and $n = 58$ for Study 4.

When exploring the moderating effect of vignette valence, we found a significant interaction, $F(1, 159) = 12.89$, $p < 0.001$, $partial\ \eta^2 = 0.08$, 95% CI = [0.02, 1.00], such that the AI responses were rated as especially more compassionate than human responses when the empathy prompts contained a negative experience ($B = 0.85$, SE = 0.104, $p < 0.001$) than when they contained a positive experience ($B = 0.32$, SE = 0.10, $p = 0.002$). The summarized findings for the effect of vignette valence for the latter and subsequent studies can be found in Fig. 4.

**Experiment 2**

Experiment 2 replicated the main findings of experiment 1. We found a main effect for empathy source, $F(1, 195) = 63.18$, $p < 0.001$, $partial\ \eta^2 = 0.24$, 95% CI = [0.16, 1.00], such that the GPT-generated responses (M = 4.06, SD = 0.65) were rated as more compassionate than the select human-generated responses

(M = 3.60, SD = 0.63). However, we also found an interaction between empathy source and the transparency condition, $F(1, 195) = 10.84$, $p < 0.001$, $\eta^2 = 0.05$, 95% CI = [0.01, 1.00], indicating that AI's empathy advantage was larger when participants were blind to the empathy source ($B = 0.65$, SE = 0.08, $p < 0.001$). Despite this interaction, participants still rated AI as more compassionate even when AI was transparently labeled ($B = 0.27$, SE = 0.08, $p < 0.001$). AI responses, in other words, were clearly rated as more compassionate than humans' even when participants knew the response was generated by AI.

We also examined whether response valence moderated the interaction between author and condition. Although we did not find a significant 2 x 2 x 2 interaction, $F(1, 585) = 0.39$, $p = 0.53$, we did find an interaction between author and valence, $F(1, 585) = 30.70$ $p < 0.001$, $partial\ \eta^2 = 0.05$, 95% CI = [0.02, 1.00], such that the difference between AI and human responses was greater for the negative scenarios ($B = 0.67$, SE = 0.05,

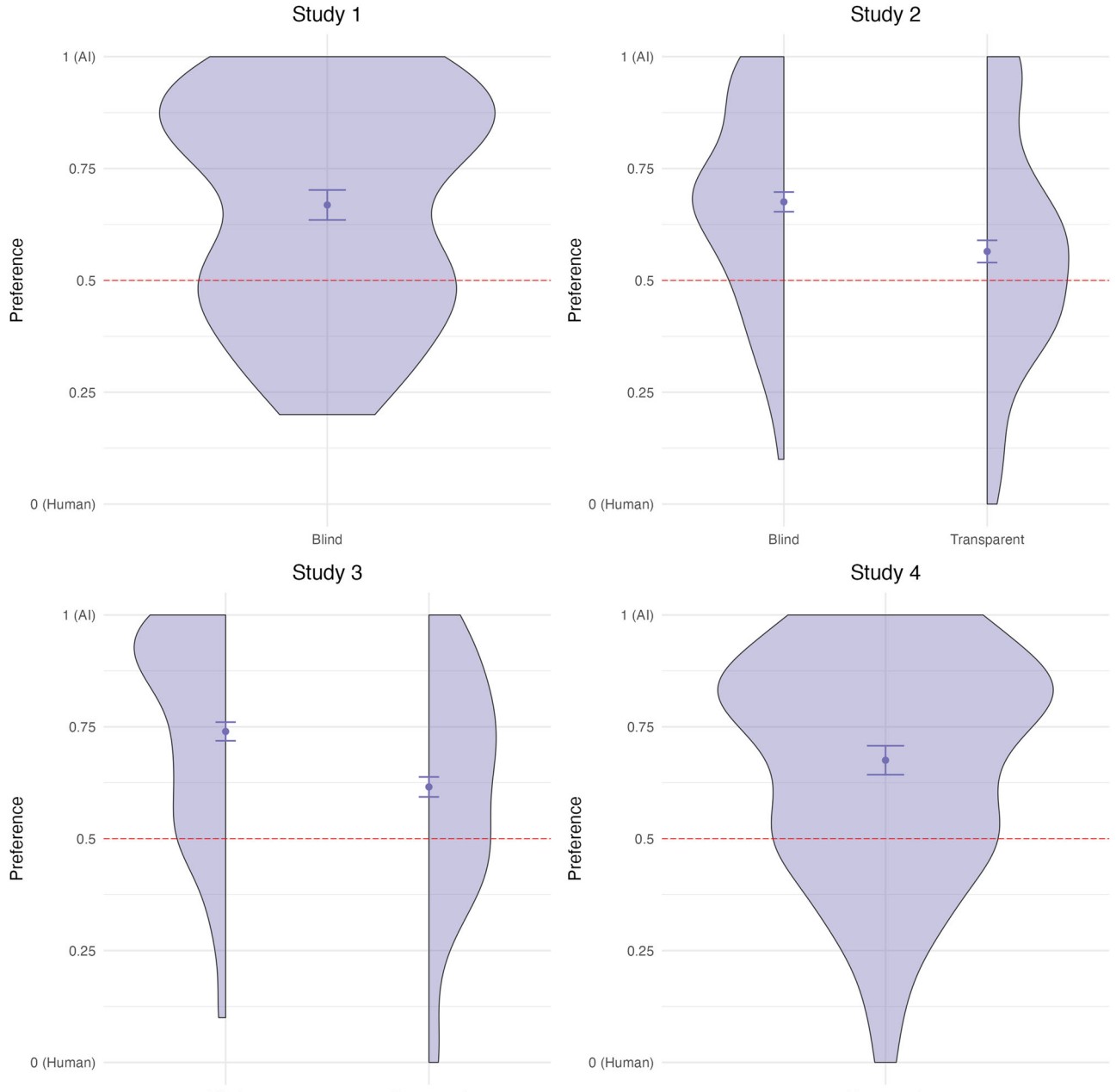

**Fig. 3 | Preference ratings for AI versus human-authored responses across studies under different transparency conditions.** Violin plots illustrate preference distributions where 0 denotes a preference for human-authored responses and 1 denotes a preference for AI-authored responses, across four separate studies. Panels represent Study 1 through 4, segmented into conditions where the response author labels were concealed or transparent to evaluators. The dotted red line at 0.5 indicates no preference for human or AI responses. Error bars denote 95% confidence intervals. The sample sizes are $n = 54$ for Study 1, $n = 197$ for Study 2 ($n = 98$ blind, $n = 99$ transparent), $n = 247$ for Study 3 ($n = 126$ blind, $n = 121$ transparent), and $n = 58$ for Study 4.

$p < 0.001$) than for the positive scenarios (B = 0.24, SE = 0.05, $p < 0.001$). This suggests that AI had a greater advantage over humans when addressing vignettes describing negative experiences, and this was the case whether the empathic responses were transparently labeled or not.

Finally, we also examined response preference. We found that the AI responses were judged as better at addressing the prompt than the select human responses, $t(196) = 7.04$, $p < 0.001$, d = 0.50, 95% CI = [0.35, 0.65]. We further found significant differences in participants' response preferences by transparency condition, such that the preference for AI-generated responses was greater when participants were blind (M = 0.68, SD = 0.22) rather than transparently exposed (M = 0.57, SD = 0.25) to the response author labels, $t(195) = 3.34$, $p = 0.001$, d = 0.48, 95% CI = [0.19, 0.76].

**Experiment 3**

Experiment 3 had a design like experiment 2, except the human responses were created by trained hotline crisis responders. Again, we found a main effect of author, $F(1, 245) = 154.36$, $p < 0.001$, *partial* $\eta^2 = 0.39$, 95% CI = [0.31, 1.00] such that AI responses (M = 4.08, SD = 0.63) were rated as more compassionate than human responses (M = 3.47, SD = 0.60). As with Experiment 2, however, this main effect was subsumed under a significant interaction between response source and the transparency condition, $F(1, 245) = 20.81$, $p < 0.001$, *partial* $\eta^2 = 0.08$, 95% CI = [0.03, 1.00], suggesting that AI's compassion advantage over expert humans was stronger when participants were blind to author source. Nonetheless, simple effects analyses indicate that AI's responses were rated as more compassionate than

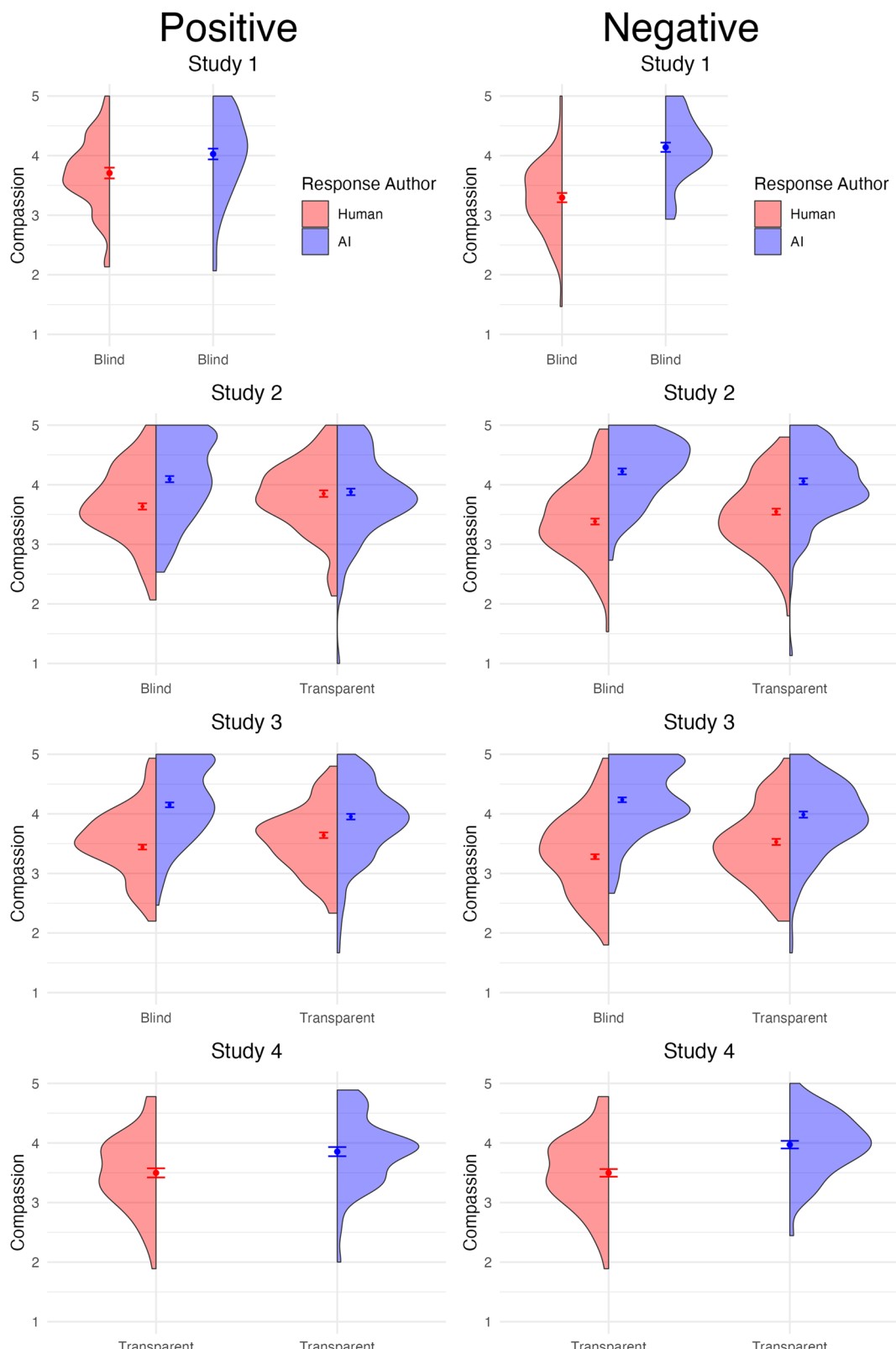

**Fig. 4 | Compassion ratings by valence and transparency condition across four studies.** Violin plots display compassion ratings for AI versus human-authored responses, split by positive (left column) and negative (right column) vignette valence across four studies. Error bars denote 95% confidence intervals. The sample sizes are $n = 54$ for Study 1, $n = 197$ for Study 2 ($n = 98$ blind, $n = 99$ transparent), $n = 247$ for Study 3 ($n = 126$ blind, $n = 121$ transparent), and $n = 58$ for Study 4.

humans in both the blind (B = 0.83, SE = 0.07, $p < 0.001$) and transparent (B = 0.38, SE = 0.07, $p < 0.001$) conditions.

In experiment 3, we again did not find a 2 x 2 x 2 interaction between response author, condition, and valence, $F(1, 735) = 0.53$, $p = 0.47$, but we replicated the author by valence interaction from study 2, $F(1, 735) = 8.37$, $p = 0.004$, *partial* $\eta^2 = 0.01$, 95% CI = [0.00, 1.00], such that perceived compassion was even greater for AI than expert humans when it addressed negative prompts (B = 0.71, SE = 0.05, $p < 0.001$) than when it addressed positive prompts (B = 0.51, SE = 0.05, $p < 0.001$).

Response preference also followed the pattern of the two previous studies, such that AI responses were considered better at addressing the prompt than expert humans, $t(246) = 11.38$, $p < 0.001$, d = 0.72, 95% CI = [0.58, 0.86]. We once again found significant differences in participants' response preferences by transparency condition, such that the preference for AI-generated responses was greater when participants were blind (M = 0.74, SD = 0.23) rather than transparently exposed (M = 0.62, SD = 0.25) to the response author labels, $t(245) = 4.06$, $p < 0.001$, d = 0.52, 95% CI = [0.26, 0.77]. In examining the extent of participants' preferences for compassionate statements generated by AI v. human experts separately across blind and transparent conditions against a test value of 0.5, we confirmed that AI responses were judged as better at addressing the prompt than the expert human responses to a greater extent in the blind, $t(125) = 11.5$, $p < 0.001$, d = 1.02, 95% CI = [0.81, 1.24], rather than transparent condition, $t(120) = 5.18$, $p < 0.001$, d = 0.47, 95% CI = [0.28, 0.66].

### Experiment 4

Experiment 4 used some of the same human expert and AI responses as Experiment 3, but all the responses were transparently labeled. In addition, participants rated how understanding, validating, and caring each response was. Regarding responsiveness, we hypothesized that AI responses would be rated as expressing greater responsiveness than empathic responses generated by expert humans (crisis line workers). Specifically, we hypothesized that AI responses would be rated as more understanding, validating, and caring.

As with the earlier three studies, AI-generated responses were rated as significantly more compassionate (M = 3.91, SD = 0.47) than the expert human-generated responses (M = 3.41, SD = 0.51), $F(1, 57) = 32.69$, $p < 0.001$, *partial* $\eta^2 = 0.36$, 95% CI = [0.20, 1.00]. Similarly, AI responses were preferred to expert human responses with respect to being better at addressing the prompt, $t(57) = 5.41$, $p < 0.001$, d = 0.71, 95% CI = [0.42, 1.00].

We further replicated the interaction effect between author and valence, $F(1, 171) = 4.00$, $p = 0.04$, *partial* $\eta^2 = 0.02$, 95% CI = [0.00, 1.00], with simple effects analyses suggesting that compassion ratings for AI were greater when it addressed negative prompts (B = 0.64, SE = 0.10, $p < 0.001$) than when it addressed positive prompts (B = 0.36, SE = 0.10, $p = 0.002$).

Finally, AI-generated responses (M = 3.24, SD = 0.47) were rated as significantly more responsive than the expert human-generated responses (M = 2.97, SD = 0.51), $t(57) = 4.57$, $p < 0.001$, d = 0.60, 95% CI = [−0.88, −0.32]. Specifically, AI responses outperformed human experts across all three facets of partner responsiveness: AI responses were evaluated as significantly more understanding (M = 3.23, SD = 0.48 vs. M = 2.99, SD = 0.53), $t(57) = 3.86$, $p < 0.001$, d = 0.51, 95% CI = [0.23, 0.78]; validating (M = 3.54, SD = 0.45 vs. M = 3.24, SD = 0.52), $t(57) = 4.49$, $p < 0.001$, d = 0.59, 95% CI = [0.31, 0.87]; and caring than expert human responses (M = 3.24, SD = 0.52 vs. M = 2.69, SD = 0.68), $t(57) = 7.78$, $p < 0.001$, d = 1.02, 95% CI = [0.70, 1.34]. These findings highlight that AI is not only perceived as broadly responsive but also surpasses human experts in demonstrating understanding, validation, and care.

After further examining responsiveness, we found a significant author by valence interaction, $F(1, 171) = 4.00$, $p = 0.047$, *partial* $\eta^2 = 0.02$, 95% CI = [0.00, 1.00]. Post-hoc comparisons suggested that responsiveness ratings for AI were greater when it addressed negative circumstances, (B = 0.64, SE = 0.10, $p < 0.001$) than positive circumstances (B = 0.36, SE = 0.10, $p = 0.006$).

Finally, to explore whether responsiveness ratings mediated the effect of author on compassion, we conducted a within-subjects mediation analysis. The indirect effect of empathy source on compassion ratings through responsiveness was significant, as the bootstrap confidence interval based on 1000 samples did not include zero, 95% CI = [0.1823, 0.3923], $p < 0.001$. Furthermore, analysis of the direct effect revealed that response author still significantly predicted compassion ratings even after accounting for responsiveness, $F(1, 113) = 18.04$, $p < 0.001$, indicating partial mediation. Responsiveness itself was a strong predictor of compassion ratings, $F(1, 113) = 108.12$, $p < 0.001$. These results suggest that while part of why AI responses are rated as more compassionate is that they are perceived as more responsive, perceived responsiveness cannot explain all the variance in compassion ratings for AI.

### Discussion

Empathy has numerous benefits on its recipients, but the toll associated with its expression, with competing pressures[12,21-23], can facilitate avoidance and a reduction in prosocial behaviors[4-7]. The gap between empathic supply and demand leaves recipients with unfulfilled needs for supportive care and contributes to heightened reports of social isolation, loneliness, and mental health concerns[18,35,36], which have intensified since the COVID-19 pandemic, particularly among youth[35]. Given these challenges, researchers have examined whether Artificial Intelligence (AI) would be perceived as comparably effective to humans in providing empathic support[16-20]. Recent research suggests that AI can indeed be effective in promoting healthy social behaviors[37] like self-disclosure[19], trust, enjoyment, intimacy[24], and improved mood[18].

In the present study, we asked if third party evaluators would rate responses made by AI as more compassionate than responses made by select and expert humans. Across four preregistered experiments, the results robustly supported our initial hypotheses: AI-generated responses were consistently rated as more compassionate and were preferred over human-generated responses. Though AI may not express authentic empathy or share others' suffering[15], it can express a form of compassion through its facilitation of active support[8]. In fact, it is so effective that third-party evaluators perceive it as being better than skilled humans. While AI does not experience empathy in the psychological sense, it is important to note that empathy is an interaction between two entities, rather than solely an internal experience of the empathizer. From this perspective, the interacting partner could still derive the benefits of empathic engagement, even when it originates from an artificial system. In the present study, the perception of AI's empathic responses might bring about effects in its recipients that could be similar to (or even better than) the effects of empathy expressed by humans —at least through the eyes of third-party evaluators.

In experiments 2 and 3, we randomly assigned participants to blind and transparent conditions, where they were ignorant and aware of the response author identity, respectively. In an aim to assess the AI advantage against the influence of source transparency, we confirmed that the AI advantage decreased when people knew the response authors' identities. However, participants continued to rate AI responses as more compassionate than human responses even when they knew that the response was AI-authored. This pattern of findings is consistent with the existing literature surrounding the impact of source disclosure on people's perceptions of AI-generated content, where the effectiveness of AI-generated content is lower when source identity is disclosed, even when its quality is evaluated as largely comparable to human responses[20,38,39]. In short, people seem to prefer AI content more when they are unaware that it was made by AI.

One partial explanation for these findings is offered by the literature surrounding algorithm aversion (or rather, human favoritism)[29]. Literature surrounding algorithm aversion posits that knowing that a piece of content is AI-generated biases its reception, though reactions vary with AI's contextual application[29,40]. Alternatively, the devaluation of AI-generated content may come from legitimate skepticism about AI's capabilities in the context of empathy, given its inability to embody genuine feeling or care[15]. While AI aversion is common[40,41], it can be partly overcome with

experience[42,43], successful use[29], and framing that emphasizes AI's supportive motives[44]. Taken together, these observations suggest that negative initial impressions of AI effectiveness can improve as individuals gain more experience and positive outcomes with AI.

Following these findings, one potential future direction lies in asking whether this is true for all people or only a subset. People's expectations of AI largely depend on their perception of it—shaped by numerous attitudinal, social, and cognitive variables—which can impact the perceived value and experience of AI-generated support[20,26,27,44]. Thus, it is crucial to examine the heterogeneity of these findings.

Notably, we established that AI responses maintained their compassion advantage even when compared to those from a subset of crisis line responders, trained experts in delivering empathic support to a diverse Canadian population. This advantage persisted across both blind and transparent source identity conditions, including when all participants were aware of the authors' identities for each empathic response (Study 4). These findings are particularly significant since responders in this organization undergo extensive training before selection[45] and may work concurrently in health fields that tend to centralize empathic communication with clients[12,22,46]. Despite their training, these individuals report constraints like time pressures, high-severity cases[22], and competing demands, which can contribute to compassion fatigue, staff shortages, and diminished client care and trust[22,23]. Given these factors, a sample of regular individuals selected for their empathic abilities could perform as well as, or even surpass, crisis line workers in delivering empathic responses.

Further support for the notion that external constraints on professional human empathizers may limit the quality of their empathic communication was offered by a supplemental comparison of expert to non-expert (select) Prolific responders, whose carefully selected responses reflected high quality of content, emotional salience, and relatability to each scenario in the vignette. A detailed report of this evaluation can be found in Supplementary Note 2 of the Supplementary Information file, which revealed that our select responses were rated as no less compassionate than experts' by third-party raters, and neither authored response was preferred over the other. In sum, despite the high quality of empathic responses from both samples of human responders, AI's consistent performance in providing superior empathetic responses highlights its utility in compassionate care through complementing, or potentially enhancing, human communication[8] and preparedness, particularly in brief, written contexts. This utility is underscored by recent findings demonstrating that support workers can successfully use AI collaboratively to guide their empathic responses[46].

The overall observation that AI-generated responses were rated as more effective than those produced by trained empathic professionals challenges conventional assumptions regarding human expertise and highlights the difficulty in overcoming the costs and constraints associated with the expression of empathy[16,47]. In contrast, AI consistently provides empathic support without showing a decline in empathy quality[16] or context-appropriate responding[44]. This advantage may contribute to AI's sustained high ratings for responsiveness[30] in the present study, which partially explains its greater perceived compassion relative to both expert and non-expert human responders. Specifically, AI responses were rated as more understanding than human responses, as they actively solicited more details, summarized, and expressed understanding. AI responses were also rated as expressing more validation through their greater acknowledgment of the individual's feelings and use of expressive language. In terms of caring, AI responses were evaluated to significantly outperform humans' by more effectively expressing empathy and support and engaging more deeply with the hypothetical individuals' experiences[30].

Our results further indicate that AI had an advantage over humans when responding to negative prompts, such as in vignettes that depicted suffering and sadness. Interestingly, while AI was also perceived to be better than select and expert humans at responding to positive prompts depicting joy, this advantage was not as apparent. Why might that be? One rationale may be that familiarity with one's social partner is particularly important for the expression of empathic support under positive rather than negative circumstances and is expressed more readily to close others than strangers[48]. An additional explanation for the observed differences is that expressed compassion (empathic concern), distinctively aimed at alleviating distress[8,49], may have heightened the salience of AI's responses to typical negative prompts, as people generally associate empathy with responses to pain or suffering.

Cumulatively, these findings highlight the communicative skill and value of generative AI systems like ChatGPT and have profound implications for further integration of AI in domains requiring expressed empathy. Public perception of AI and its involvement in empathic support is complex, influenced by diverse individual and perceptual factors[19,20,26,27]. Nevertheless, the consistent preference for AI-generated responses in the present study, even when compared to trained professionals and varied transparency conditions, suggests a significant shift in how AI's role in communication might be perceived and potentially managed in the future, particularly in areas demanding consistent, high-quality exchanges.

In addition to the strengths of empathic AI, it is nevertheless important to note the prospective risks of its empathic expressions for both the recipients and human providers of empathy[16]. In particular, there are highlighted ethical concerns surrounding non-transparent AI use in empathy delivery, emphasizing the need for informed recipient choices regarding the source from which they obtain supportive care[16]. Moreover, an overreliance on empathic AI may increase the demands for personalized and unconditional support from recipients, which could undermine existing human effort, reinforce problematic behavior, and contribute to a counterproductive increase in mental health concerns[16]. Thus, a balanced approach that leverages both AI and human communicative strengths is essential, ensuring that the integration of empathic AI fosters positive change while respecting ethical standards and supplementing, rather than replacing, human-based care.

## Limitations

While the results across the present four studies are promising, several limitations should be noted. First, given that the AI and human-generated responses were rated by third-party evaluators, the patterns of findings may not generalize to interactions in which the evaluators are direct recipients of empathy. Future research could assess whether AI's advantage in providing empathic support relative to fellow humans maintains and informs participant preferences through direct recipient feedback[20]. Additionally, the present study did not examine whether familiarity with and use of AI technology may have differentially influenced evaluations of AI and human-generated responses across blind and transparent conditions. Familiarity and proficiency with AI, among other personality and social variables, play an important role in shaping attitudes towards AI[19,20,24,26,27]. Given that the present study and related efforts to assess empathic AI relative to humans have done so through brief interactions[16-20], future studies need to examine more long-term interactions with empathic AI to establish whether people's preferences and attitudes towards AI change as a function of time and assess the role of empathic AI in supporting experienced users' mental health[50].

## Conclusions

In sum, our study demonstrates the strengths of AI in communication contexts that require empathic expressions, albeit from a third-party lens. Participants consistently rated AI-generated responses as more compassionate, understanding, validating, and caring; they knowingly preferred AI responses to human-generated responses when author identity was made explicit and even when the human comparison was comprised of trained empathy first responders. Ultimately, AI's ability to consistently deliver compassionate communication positions it as a strategic asset in support scenarios where human resources are stretched thin.

## Data availability

All datasets and materials are available at the repository https://osf.io/wjx48/.

## Code availability
Code is available at the repository https://osf.io/wjx48/.

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

## Acknowledgements

We would like to thank Gregory Depow, Leif Anderson, Katy Tam, Dasha Sandra, Yiyi Wang, and all other members of the Work and Play Lab for their support in material validation and insights along the way.

## Author contributions

D.O. and M.I. conceived the study design and research questions. D.O. programmed the experimental iterations, collected, and cleaned the data. D.O. and V.O.M. analyzed the data. D.O. wrote the manuscript. M.I. and V.O.M. edited the manuscript and provided crucial oversight of and feedback on the work.

## Competing interests

The authors declare no competing interests.
