## [Transparent Peer Review file · Communications Psychology]

Third-Party Evaluators Perceive AI as More Compassionate than Expert Humans

Corresponding Author: Professor Michael Inzlicht

Version 0:

Decision Letter:

Dear Professor Inzlicht,

Thank you for your patience during the peer-review process. Your manuscript titled "The Kindness Machine: AI Outperforms Expert Humans in Expressing Compassion" has now been seen by 3 reviewers, and I include their comments at the end of this message. They find your work of interest but raised some important points. We are interested in the possibility of publishing your study in Communications Psychology, but would like to consider your responses to these concerns and assess a revised manuscript before we make a final decision on publication.

We therefore invite you to revise and resubmit your manuscript, along with a point-by-point response to the reviewers. Please highlight all changes in the manuscript text file.

Editorially, the key priority is that you present the aims and achievements of the study more closely to the data. It should be clear throughout the text (and the title) that the study measures "perceived" or "rated" empathy, as judged by third-party raters. While we do not ask for a reframing of the study, we expect that critical relevant literature is cited and that it becomes clear how your work confirms, extends, or contradicts previous accounts. We further ask that you include in revision necessary additional analyses and better explain all methodological details that were marked as unclear by the referees.

I am attaching an Editorial Requests Table that details critical reporting requirements for the revised manuscript. Please attend to each item and ensure your manuscript is fully compliant. We are requesting that your manuscript aligns with these requirements as this facilitates the evaluation of your manuscript, reducing delays in re-review and potential future acceptance. If your revised manuscript is not aligned with these requests on major issues, such as those concerning statistics, it may be returned to you for further revisions without re-review. Additional information can be found in our style and formatting guide <https://www.nature.com/documents/commspsychol-style-formatting-guide-accept.pdf> Communications Psychology formatting guide.

Please use the following link to submit your

- revised manuscript,
- point-by-point response to the referees' comments,
- cover letter (as a separate document),
- the Editorial Policy Checklist (see below),
- the Reporting Summary (see below), and
- the completed Editorial Request Table (attached):

Link Redacted

Best regards,

Yafeng Pan

Yafeng Pan, PhD
Editorial Board Member
Communications Psychology
orcid.org/0000-0002-5633-8313

REVIEWER EXPERTISE:

Reviewer #1
Reviewer #2

REVIEWER REPORTS:

Reviewer #1 (Remarks to the Author):

The goal of this line of studies was to assess which agent was better at generating empathic statements: humans or AI. To evaluate this, the authors compared human-generated and GPT-generated written responses to empathic prompts across four preregistered experiments. This resembles similar attempts to compare human and AI-generated empathic responses (e.g., Ayers et al., 2023), which the authors acknowledge. However, the current paper takes this comparison one step further, both by comparing AI responses not only to laypeople but also to professional empathizers, and by examining the effect of disclosing (versus not disclosing) the identity of the recipient (AI/human). Both are important contributions to the field and can assist in answering questions regarding the use of AI in situations where empathy, compassion, and emotional support are needed. The authors hypothesized that, in all cases, AI responses would be preferred over human responses (binary choice) and rated as more empathic and compassionate than human responses. They preregistered all hypotheses.

The paper is well-written, interesting, and easy to follow. I enjoyed reading it!

The main limitation of the study (which is acknowledged in the discussion) is that it does not study real interactions with an AI or human. Instead, similar to Ayers et al., it uses third-party raters to assess empathy, compassion, and responsiveness. Examining the actual effect of such interactions on the person sharing their emotional event would make their point stronger. If the authors decide not to run an additional study with real interactions, they should tone down the conclusiveness in the discussion. The fact that third-party raters prefer the AI response does not mean that support-seekers would prefer this as well.

The introduction is clearly written, easy to follow, and summarizes the main papers in the field.

Results: Results are clearly written and easy to follow. They are also highly consistent, showing that AI responses are rated (by a third party) as more empathic, compassionate, and responsive. The authors also consistently find an interaction between response source and transparency, such that when the AI identity is revealed, the effects are smaller but still significantly different from human responses.

Discussion:

The authors make a strong claim regarding the benefits of empathic and compassionate responses generated by AI. However, since the responders were always third-party raters, I think this point should be toned down. The authors do acknowledge this limitation, but I think this point should also make the discussion of the current results more nuanced. While third-party raters rate the AI responses as more empathic, it is not clear that first-person users will actually prefer them or choose to continue talking to them rather than to a human or support caller when given the choice. This should be more clearly stated.

Moreover, the authors refer to numerous biases regarding AI that can shape people's perceptions of support, potentially undermining its actual effectiveness and perceived value, and to AI aversion and human favoritism. While this is true for some aspects of cognition, I am not sure that I would call it a bias when referring to empathy or compassion. Wouldn't it be a realistic understanding (rather than a bias) to consider that ChatGPT does not really feel or care? In this sense, it makes

sense to reduce the ratings of feelings of empathy and compassion when you know that the response was from an AI that does not genuinely care. I suggest highlighting this point in the discussion rather than referring to this as a bias.

Reviewer #2 (Remarks to the Author):

The authors find that people view responses generated by AI to be more compassionate than those generated by human respondents, in part because AI responses are seen as more responsive, even when participants were aware of the author of the responses.

These findings are really fascinating, and I think important. However, I found the framing around empathy and some of the descriptions of the findings to be a bit strange and problematic. In part, this reflects the strangeness of the research question itself — it is a feature of this area of study that it is simply odd to talk about AI empathizing at all. However, I think it ultimately ends up being problematic by going beyond what can actually be concluded from these findings, and claiming a contribution to a literature (i.e., the empathy literature) I'm not sure is as relevant to this work as the authors are currently claiming.

Empathy, even by the authors' definition, exists in the mind of the empathizer. It is the empathizer's ability to share an experience, take someone's perspective, and feel empathic concern. Empathy has important downstream effects on the person who is empathized with, for sure, as well as broader benefits. However, empathy, at least how I've always seen it conceptualized in the literature, is not the words or actions themselves, separate from the mind that is saying those words or dictating those actions, but the psychological experience that leads to those words and actions—something AI clearly cannot experience.

This begs the questions of: 1) How to frame this research question and its contribution (Are you really studying empathy at all?), and 2) How to precisely describe these specific findings (e.g., Is AI really "more understanding" or are its responses merely perceived as more understanding because of certain linguistic cues it is using?).

With respect to the first concern, perhaps you could frame this work as studying the effectiveness of empathic AI, and talk about it in terms of its contribution to the literature on human reactions to AI generated content, rather than its contributions to the psychology of empathy. You could still bring in psychological literatures—including the literature on empathy—that highlight the need for humans to feel heard and validated and cared for, but without making the leap to claiming that you are actually studying the psychology of empathy in these studies. In other words, make it more like a human-computer interaction paper.

With respect to the second concern, I think you want to be super precise about how you refer to these findings by talking about participants' assessments of empathy and understanding, rather than actual empathy and understanding. For example, I took issue with the phrase (from p. 16), "AI responses were more understanding than human responses, by actively soliciting more details, summarizing, and acknowledging understanding." However, I would be okay with a phrase such as, "AI responses were rated as more understanding than human responses due to the fact that they actively solicited more details, summarized, and claimed understanding." There are many such phrases throughout the paper.

Lastly, I had some trouble following the experiments given the formatting of the methods and results, however, if I understand Experiment 2 correctly, there is no condition in that study (or elsewhere) where humans use AI to generate a statement, correct? It seems to me that AI is doing a better job of generating the specific words that people want to hear—in part because it's just mirroring what someone says. So, an important question in my mind is what would happen if humans used AI to generate the perfect response, in essence combining the two conditions. That is, what would happen if it was clear to participants that a human was using an AI generated response.

In sum, I think this is a super interesting question and a fascinating set of findings. However, I think you want to make sure you're not jumping the shark to ensure you are being true to what you can really conclude here, and so this work gets an appropriate and productive reception both academically and in the popular media.

Reviewer #3 (Remarks to the Author):

There is a lot to like in "The Kindness Machine: AI Outperforms Expert Humans in Expressing Compassion". The research contributes to the debate, by comparing how third parties judge AI and expert generated responses in regard to their compassion.

There is a lot of value in comparing the responses with experts, rather than layman, as in the context chosen, AI is suggested as a supplement or even as an alternative to experts, rather than of friendly layman advice.

I have some questions regarding the overall framing, which I believe might profit from some tweaking, and a very basic additional analyses to suggest – all suggestions which I believe can be well addressed in a revision.

A second named contribution is the source disclosure of the AI. I think this area is actually very well studied, especially in adjunct fields, such as communications and marketing, as well as computer science proceedings. I myself reviewed three different papers on similar topics, with every single one of them manipulating source disclosure. I believe this is important to do, but framing wise would focus a bit more on the expert element as a main contribution (for source disclosure you could relate it more to how it fits the existing literature, e.g. compared with much other work you seem to find less of a neg. effect after disclosure).

The focus of the paper is on empathy, with AI outperforming humans - a quick reader might take away that we can just rely on the AI in clinical/therapy contexts now. I think it is important to make explicit, that the main goal of a medical professional is not always just being empathetic, and sometimes is not even the main goal, compared with e.g. treating the patient in an emergency situation. I am not at all suggesting empathy is not crucial, but I think one just needs to be careful in overall framing, as showing equivalence in empathy is a first, but very early step, with this research not aiming to evaluate if the given responses would actually lead to any desirable behaviour changes/outcomes etc..

You mention that the LLM normally generate longer responses, and give as a potential reason the taxing nature of humans expressing empathy. This seems to be an empirically unsupported claim, and one that I would not expect to be the primary driver myself, having exploratively interviewed medical students and professionals in the past: short responses are probably much more driven by time constraints, and general demand, rather than by a lack of empathy from the provider, or an exhaustion of their empathy as they go through their day. Similarly, it was suggested that shortage of mental health workers is actually linked to fatigue of empathy, while this could be a partial reason, it seems a bit too strong of a claim to me, as it is currently presented as the main driver.

Another framing related point, that I think you briefly touch on, the interactions are very short term. I believe this does not make them comparable to real therapy, but maybe more of a brief comforting situation. Therapy involves long term, contentious interactions. We are seeing AI's memory increasing at the moment – so I am not suggesting that similar results shouldn't be expected in therapy, but the study context for me is closer to the crises phone worker, than e.g. typical therapy. There is one sentence along these lines in the discussion, but I think being more explicit, and spending some more depth on here would be important for readers to be able to contextualise the work appropriately, as well as giving you the chance to guide future research.

One methodological concern, that I don't think could be solved easily: Your Study 1 and 2 samples are the result of picking the most empathetic out of a group, in Study 3 you use experts, i.e. crises workers, assuming their responses will be better. This assumption seems to be unfounded to me, while I expect a randomly chosen participant to not outperform an expert, I actually expect the best out of a decently seized sample to outperform an expert. As we don't evaluate just outperforming, but rather outperforming in empathy specifically, the prediction gets more muddy to me, as an expert might follow other goals, too.

Related, I would appreciate some more details for the methods, e.g. did the same participants do the prompts and the selection? How was it chosen which ones are most empathetic? What was the sample size of prompts, from which the best 10 were chosen etc.. Were groups instructed to be empathetic?

One additional analyses I believe could be useful in this context, is to have some between study comparisons. If I understand the set-up correctly, you have the same statements being judged partially in the skilled layman and in the expert condition, which means you could check the underlying assumption that experts are actually sig. better than selected layman at empathy.

On a more minor note, I found the entire article very pleasant to read and understandable, with the only exception being the abstract. Not having read the paper, I was confused in what would actually differentiate the conditions. Condition 1 is presented as skilled human responders, which I took to be as experts, but Condition 3 turned out to be the experts. Study 4 was then introduced as "transparency was kept constant", which I found also a bit confusing at first. (Again it all made sense once I had read the introduction and paper, but was just a bit more difficult when I first read the abstract). Maybe some work could be undertaken here to make a reader, who is not familiar with the context or condition, aware of what each study is adding at-large.

On a very minor note, you mention that the average effect size in social psychology is .4? I remember more recent work, e.g. by Funder having found less than half, but might be mistaken.

Overall, I believe that the paper is interesting and is in a highly relevant context, and I am looking to read the revised version, for which I hope these comments are helpful.

Best wishes,

Andrés Gvirtz (Note: While I normally do not sign my reviews, I note that the journal actively encourages reviewers to do so, and therefore complied)

EDITORIAL POLICIES

We ask that you ensure your manuscript complies with our editorial policies and reporting requirements.

To that end, we require revised manuscripts to be accompanied by two completed items: a reporting summary that collects information on study design and procedure, and an editorial policy checklist that verifies compliance with all required editorial policies.

- <https://www.nature.com/documents/nr-reporting-summary.zip>>Nature Research Reporting Summary
- <https://www.nature.com/documents/nr-editorial-policy-checklist.pdf>>Editorial Policy Checklist

All points on the policy checklist must be addressed. Your revised manuscript can only be sent back to the referees if these checklists are completed and uploaded with the revision.

Notes: If you have submitted a Stage 1 Registered Report, Review, Primer, Comment, or Perspective you do not need to submit these forms. If you have already submitted these forms, you may disregard this request.

** Visit Nature Research's author and referees' website at <http://www.nature.com/authors>>www.nature.com/authors for information about policies, services and author benefits**

If you experience problems in linking your ORCID, please contact the <http://platformsupport.nature.com/>>Platform Support Helpdesk.

Version 1:

Decision Letter:

Dear Mike,

Your manuscript titled "The Kindness Machine: Third-Party Evaluators Perceive AI as More Compassionate than Expert Humans" has now been seen by our reviewers, whose comments appear below. In light of their advice I am delighted to say that we are happy, in principle, to publish a suitably revised version in Communications Psychology.

We therefore invite you to revise your paper one last time to address the remaining concerns of our reviewers and a list of editorial requests. At the same time we ask that you edit your manuscript to comply with our format requirements and to maximise the accessibility and therefore the impact of your work.

EDITORIAL REQUESTS:

SUBMISSION INFORMATION:

OPEN ACCESS:

Link Redacted

Best wishes,

Marika & Yafeng Pan

Marika Schiffer, PhD
Chief Editor
Communications Psychology

REVIEWERS' COMMENTS:

Reviewer #1 (Remarks to the Author):

I am satisfied with the authors' response and am looking forward to seeing this paper published.

Reviewer #2 (Remarks to the Author):

I thank the authors for their conscientious response to my comments in the last round. All of my concerns have been addressed.

Reviewer #3 (Remarks to the Author):

I thank the authors for the thorough, collaborative manner in which they addressed the previous round's comments. The revisions demonstrate a strong engagement with the feedback, particularly when it came to refining the framing, contextualising the study slightly better within what is already known, and clarifying the methodological details. I also appreciate the additional study and analyses.

The contribution is now slightly narrower, but I believe it is a better representation of the findings and implications, which I believe both to be meaningful. The 10 comments I previously had have all been addressed in a comprehensive manner, with the 7 page response to them being indicative of the work that went into addressing them. Overall, an interesting study, that I am confident will add to the literature!

October 23, 2024

Dear Reviewers

We would like to express our sincere gratitude for your thorough and insightful feedback on our manuscript, now titled "The Kindness Machine: Third-Party Evaluators Perceive AI as More Compassionate than Expert Humans" (Manuscript ID COMMSPSYCHOL-24-0351-T), under consideration at *Nature Communications Psychology*. My co-authors and I greatly appreciate the time and effort you dedicated to reviewing our work. Your thoughtful comments and suggestions were invaluable in guiding us to improve the manuscript, and we have implemented the necessary changes with careful attention to the concerns you raised.

Below, I present a summary of the point-by-point revisions and associated responses to the feedback made by each reviewer. I interweave all review points with our responses, taking care to point out the specific areas of the manuscript where the changes were implemented. I have also highlighted the parts of the manuscript in yellow where the edits and any additional corrections were made to ease your review of our manuscript.

Reviewer 1 Comments

1. "The goal of this line of studies was to assess which agent was better at generating empathic statements: humans or AI. To evaluate this, the authors compared human-generated and GPT-generated written responses to empathic prompts across four preregistered experiments. This resembles similar attempts to compare human and AI-generated empathic responses (e.g., Ayers et al., 2023), which the authors acknowledge. However, the current paper takes this comparison one step further, both by comparing AI responses not only to laypeople but also to professional empathizers, and by examining the effect of disclosing (versus not disclosing) the identity of the recipient (AI/human). Both are important contributions to the field and can assist in answering questions regarding the use of AI in situations where empathy, compassion, and emotional support are needed. The authors hypothesized that, in all cases, AI responses would be preferred over human responses (binary choice) and rated as more empathic and compassionate than human responses. They preregistered all hypotheses."

We are glad that the reviewer was clearly able to understand our study and reiterate its contribution to extending the comparison of AI generated empathic responses to professional empathizers. We further thank the reviewer for recognizing the value of our findings.

2. "The paper is well-written, interesting, and easy to follow. I enjoyed reading it!"

We thank the reviewer for their kind words and positive feedback and are pleased that they found the paper enjoyable and easy to follow.

3. “The main limitation of the study (which is acknowledged in the discussion) is that it does not study real interactions with an AI or human. Instead, similar to Ayers et al., it uses third-party raters to assess empathy, compassion, and responsiveness. Examining the actual effect of such interactions on the person sharing their emotional event would make their point stronger. If the authors decide not to run an additional study with real interactions, they should tone down the conclusiveness in the discussion. The fact that third-party raters prefer the AI response does not mean that support-seekers would prefer this as well.”

We thank the reviewer for this very important point and agree that the interpretation of our findings should be toned down given our use of third-party ratings, rather than direct recipient feedback, to compare the empathic quality of AI and human responses. As we decided to not run additional study iterations with direct recipients for this specific paper, we took care to reframe our discussion around the effectiveness of AI in generating responses that could be *perceived* as expressing empathy relative to human responses, at least from a third-party standpoint and with respect to brief, written communication. We further qualified language surrounding AI’s relative benefits throughout the manuscript to include words like “perceived compassion”, “perceived responsiveness”, “perceived effectiveness”, “expressed compassion”, “communicative strengths”, to avoid extrapolating any strengths of empathic responses from AI from our third-party sample to direct communication. An example of further change can be seen on **page 17** in the discussion section, where we now write,

“Though AI may not express authentic empathy or share others’ suffering¹⁵, it can express a form of compassion through its facilitation of active support⁸. In fact, it is so effective that third-party evaluators perceive it as being better than skilled humans. In other words, while AI does not experience empathy in the psychological sense, it is important to note that empathy is an interaction between two entities, rather than solely an internal experience of the empathizer. From this perspective, the interacting partner could still derive the benefits of empathic engagement, even when it originates from an artificial system. In the present study, the perception of AI’s empathic responses might bring about effects in its recipients that could be similar to (or even better than) the effects of empathy expressed by humans—at least through the eyes of third-party evaluators.”

Another example revision was made between **pages 19 and 20** in the discussion section, where the interpretation of third-party members’ higher compassion ratings for AI over human expert and non-expert responses was revised to

acknowledge the constraints surrounding the provision of empathic care for humans, rather than express the limits of human empathy:

“The overall observation that AI-generated responses were rated as more effective than those produced by trained empathic professionals challenges conventional assumptions regarding human expertise and highlights the difficulty in overcoming the costs and constraints associated with the expression of empathy^{16,47} In contrast, AI consistently provides empathic support without showing a decline in empathy quality¹⁶ or context-appropriate responding⁴⁴.”

In addition to the revisions on the interpretative concluding sentences for each paragraph in the discussion section, we corrected the overarching conclusion to offer a more nuanced interpretation of the results and a greater focus on the strengths of AI with respect to written communication, rather than on its broad contributions to the empathy literature. Thus, the concluding paragraph was reworded to:

“In sum, our study demonstrates the strengths of AI in communication contexts that require empathic expressions, albeit from a third-party lens. Participants consistently rated AI-generated responses as more compassionate, understanding, validating, and caring; they knowingly preferred AI responses to human-generated responses when author identity was made explicit and even when the human comparison was comprised of trained empathy first responders. Ultimately, AI's ability to consistently deliver compassionate communication positions it as a strategic asset in support scenarios where human resources are stretched thin.” (p. 22).

We believe that these and other changes that we made throughout the discussion section of our manuscript have contributed to a more careful interpretation of our findings, avoiding unsubstantiated inferences about the perceived utility of empathic expressions by AI for direct recipients of its support. We recognize and agree with the reviewer's comment regarding the strengths of examining AI's effects on direct recipients and emphasize that future research needs to investigate the first-person perspective further, as was done in a paper by Yin et al. (2024), on **page 21** in the limitations section:

“While the results across the present four studies are promising, several limitations should be noted. First, given that the AI and human-generated responses were rated by third-party evaluators, the patterns of findings may not generalize to interactions in which the evaluators are direct recipients of empathy. Future research could assess whether AI's advantage in providing empathic support relative to fellow humans

maintains and informs participant preferences through direct recipient feedback²⁰.

4. “The introduction is clearly written, easy to follow, and summarizes the main papers in the field.”

We really appreciate the reviewer’s validation of the clarity and readability of our introduction, as this is an important factor in engaging the diverse readership of the journal.

5. “Results: Results are clearly written and easy to follow. They are also highly consistent, showing that AI responses are rated (by a third party) as more empathic, compassionate, and responsive. The authors also consistently find an interaction between response source and transparency, such that when the AI identity is revealed, the effects are smaller but still significantly different from human responses.”

Again, we sincerely thank the reviewer for confirming the clarity of writing throughout the manuscript, particularly the results section, which appears to have successfully relayed the key findings across the four studies in a concise and comprehensive manner.

6. “The authors make a strong claim regarding the benefits of empathic and compassionate responses generated by AI. However, since the responders were always third-party raters, I think this point should be toned down. The authors do acknowledge this limitation, but I think this point should also make the discussion of the current results more nuanced. While third-party raters rate the AI responses as more empathic, it is not clear that first-person users will actually prefer them or choose to continue talking to them rather than to a human or support caller when given the choice. This should be more clearly stated.”

We thank the reviewer for the comment. We have changed our discussion to include statements that explore this limitation. When we explore the main findings on **page 17** (second paragraph of discussion) we qualify our findings with the statement about the third-party evaluators:

“While AI does not experience empathy in the psychological sense, it is important to note that empathy is an interaction between two entities, rather than solely an internal experience of the empathizer. From this perspective, the interacting partner could still derive the benefits of empathic engagement, even when it originates from an artificial system. In the present study, the perception of AI’s empathic responses might bring about effects in its recipients that could be similar to (or even better than) the effects of empathy expressed by humans—at least through the eyes of third-party evaluators.”

Similarly, on **page 21** in the discussion section, we now write:

“Nevertheless, the consistent preference for AI-generated responses in the present study, even when compared to trained professionals and varied transparency conditions, suggests a significant shift in how AI’s role in communication might be perceived and potentially managed in the future, particularly in areas demanding consistent, high-quality exchanges.”

This revision enabled a better specification of the utility of empathy-expressing AI in communicative settings, rather than claiming the broad benefits of AI-generated empathy. Please see our response to point 3 above for further detail on our implemented changes in result interpretation and overall framing language.

7. “Moreover, the authors refer to numerous biases regarding AI that can shape people’s perceptions of support, potentially undermining its actual effectiveness and perceived value, and to AI aversion and human favoritism. While this is true for some aspects of cognition, I am not sure that I would call it a bias when referring to empathy or compassion. Wouldn’t it be a realistic understanding (rather than a bias) to consider that ChatGPT does not really feel or care? In this sense, it makes sense to reduce the ratings of feelings of empathy and compassion when you know that the response was from an AI that does not genuinely care. I suggest highlighting this point in the discussion rather than referring to this as a bias.”

We appreciate the reviewer’s emphasized concern with this point and have revised the introduction to frame the cautious evaluation of expressions of empathy or compassion by AI as being related to a justified understanding that AI cannot emote. Moreover, we removed the word “bias” when discussing the factors that predict differential attitudes towards AI. Thus, the sentences on **page 5** were changed to the following:

“Yet, the receptiveness to AI-generated empathic responses could be influenced by the recipient’s awareness of and preconceived attitudes towards receiving support from non-human entities. For instance, people’s perception of empathy expressions from AI could be rooted in an awareness that AI, unlike humans, lacks genuine emotional experience¹⁵, and thus cannot actually care; being unmoved by empathic AI statements might then reflect warranted skepticism about its capabilities regardless of its actual effectiveness²⁰. Simultaneously, general attitudes towards AI, related to factors such as personality, conspiracy mindset²⁶, and religiosity²⁷, among others, may play a critical role in the evaluation and acceptance of AI.”

We further added a sentence on **page 5** to highlight the differences in recipients' perceptions of feeling heard after receiving human or AI-generated responses that were or were not transparently labeled, examined in a recent study by Yin et al. (2024), "*suggesting that the benefits reaped from empathic AI interactions can occur even after accounting for the drop in perceived response quality [...]*"

In **page 17** in the discussion section of the manuscript, we further supported the reviewer's point regarding the rational reduction in compassion ratings for transparently labeled, AI-generated empathic statements by confirming the consistent pattern of effects of source disclosure on individuals' perceptions of AI-generated content as such:

"In an aim to assess whether the AI advantage would hold when source identity was made transparent, we confirmed that the AI advantage decreased when people knew the response authors' identities."

"This pattern of findings is consistent with the existing literature surrounding the impact of source disclosure on people's perceptions of AI-generated content, where the effectiveness of AI-generated content is lower when source identity is disclosed, even when its quality is evaluated as largely comparable to human responses^{20,38,39}. In short, people seem to prefer AI content more when they are unaware that it was made by AI." (p. 18).

Finally, we have modified a section of the discussion section on **page 18**, which highlights the rational devaluation of empathy from AI and offers algorithm aversion as a *partial* explanation of the findings surrounding source disclosure, emphasizing that reactions towards AI-generated content vary with the context in which AI is applied. We further referenced a recent study that finds that the framing of AI as having caring intentions to interacting participants significantly influences their perceptions of its empathy (among other perceptions), even if it is not inherently caring (Pataranutaporn et al., 2023):

"One partial explanation for these findings is offered by the literature surrounding algorithm aversion (or rather, human favoritism)²⁹. Literature surrounding algorithm aversion posits that knowing that a piece of content is AI-generated biases its reception, though reactions vary with AI's contextual application^{29,40}. Alternatively, the devaluation of AI-generated content may come from legitimate skepticism about AI's capabilities in the context of empathy, given its inability to embody genuine feeling or care¹⁵ While AI aversion is common^{40,41}, it can be partly overcome with experience^{42,43}, successful use²⁹, and framing that emphasizes AI's supportive motives⁴⁴. Taken together, these observations suggest that

negative initial impressions of AI effectiveness can improve as individuals gain more experience and positive outcomes with AI.”

Reviewer 2 Comments

8. “The authors find that people view responses generated by AI to be more compassionate than those generated by human respondents, in part because AI responses are seen as more responsive, even when participants were aware of the author of the responses. These findings are really fascinating, and I think important. However, I found the framing around empathy and some of the descriptions of the findings to be a bit strange and problematic. In part, this reflects the strangeness of the research question itself — it is a feature of this area of study that it is simply odd to talk about AI empathizing at all. However, I think it ultimately ends up being problematic by going beyond what can actually be concluded from these findings, and claiming a contribution to a literature (i.e., the empathy literature) I’m not sure is as relevant to this work as the authors are currently claiming.

Empathy, even by the authors’ definition, exists in the mind of the empathizer. It is the empathizer’s ability to share an experience, take someone’s perspective, and feel empathic concern. Empathy has important downstream effects on the person who is empathized with, for sure, as well as broader benefits. However, empathy, at least how I’ve always seen it conceptualized in the literature, is not the words or actions themselves, separate from the mind that is saying those words or dictating those actions, but the psychological experience that leads to those words and actions—something AI clearly cannot experience.

This begs the questions of: 1) How to frame this research question and its contribution (Are you really studying empathy at all?), and 2) How to precisely describe these specific findings (e.g., Is AI really “more understanding” or are its responses merely perceived as more understanding because of certain linguistic cues it is using?). With respect to the first concern, perhaps you could frame this work as studying the effectiveness of empathic AI, and talk about it in terms of its contribution to the literature on human reactions to AI generated content, rather than its contributions to the psychology of empathy. You could still bring in psychological literatures—including the literature on empathy— that highlight the need for humans to feel heard and validated and cared for, but without making the leap to claiming that you are actually studying the psychology of empathy in these studies. In other words, make it more like a human-computer interaction paper.

We would like to thank the reviewer for their insightful feedback and for highlighting both the strengths and concerns regarding our framing of empathy in the manuscript. We recognize the importance of their points, particularly in the context of how AI’s capacity for empathic communication is presented and discussed, as AI does not inherently experience the dynamic process and downstream consequences of empathizing with others.

In response to these concerns, we have carefully revised the manuscript to frame the findings around the way in which AI could be leveraged in contexts where human empathizers may benefit from its support with consistent, quality written communication, and to avoid overstating AI's role in relation to human empathy. At the same time, we maintained connections to the psychological literature on empathy. While we agree with the reviewer that empathy is in the mind of the empathizer—which in the case of AI means this empathy is artificial, not real, we also submit that empathy is a dyadic process, something that exists between two agents. Thus, one can still be the recipient of empathy, even if the empathy-generating process is artificial. Hence, the need to keep the psychology in the manuscript.

Throughout the text, we took care to replace any language regarding AI's evaluated effectiveness with more precise terms such as "perceived compassion," "perceived responsiveness," "perceived effectiveness," "empathy-expressing AI," and "expressed compassion." This adjustment ensures that we are discussing AI's communicative strengths in terms of how they are perceived by third party participants, rather than implying that AI can exhibit empathy in the human sense. We further draw on the behavioral component of empathy to revise a paragraph on **page 4** in the introduction, clarifying that, "*Despite arguments that AI cannot experience empathy or feel emotions¹⁵, it can express empathy by generating responses or behaviors that appear to reflect empathic concern¹⁶ or the intention to alleviate distress⁸.*"

Similar to our response to point 3 above, we made additional revisions on **page 21** in the discussion section to clarify AI's role in generating responses perceived as compassionate and responsive:

"Public perception of AI and its involvement in empathic support is complex, influenced by diverse individual and perceptual factors^{19,20,26,27}. Nevertheless, the consistent preference for AI-generated responses in the present study, even when compared to trained professionals and varied transparency conditions, suggests a significant shift in how AI's role in communication might be perceived and potentially managed in the future, particularly in areas demanding consistent, high-quality exchanges."

9. With respect to the second concern, I think you want to be super precise about how you refer to these findings by talking about participants' assessments of empathy and understanding, rather than actual empathy and understanding. For example, I took issue with the phrase (from p. 16), "AI responses were more understanding than human responses, by actively soliciting more details, summarizing, and acknowledging understanding." However, I would be okay with a phrase such as, "AI responses were rated as more understanding than human responses due to the fact that they actively

solicited more details, summarized, and claimed understanding.” There are many such phrases throughout the paper.

Similar to point 3 above, we refined the wording and interpretation of our findings by bringing in more nuanced language to clarify the “perceived” empathic quality and responsiveness of AI-generated responses. For instance, we revise the aforementioned summary of our results for third-party raters’ evaluations of responsiveness in AI and human responses on **page 20** as such:

“Specifically, AI responses were rated as more understanding than human responses, as they actively solicited more details, summarized, and expressed understanding. AI responses were also rated as expressing more validation through their greater acknowledgment of the individual's feelings and use of expressive language. In terms of caring, AI responses were evaluated to significantly outperform humans’ by more effectively expressing empathy and support and engaging more deeply with the hypothetical individuals’ experiences³⁰.”

10. Lastly, I had some trouble following the experiments given the formatting of the methods and results, however, if I understand Experiment 2 correctly, there is no condition in that study (or elsewhere) where humans use AI to generate a statement, correct? It seems to me that AI is doing a better job of generating the specific words that people want to hear—in part because it’s just mirroring what someone says. So, an important question in my mind is what would happen if humans used AI to generate the perfect response, in essence combining the two conditions. That is, what would happen if it was clear to participants that a human was using an AI generated response.

We thank the reviewer for their thoughtful feedback and for raising the question about the potential effects of humans using AI to generate responses. We recognize the importance of exploring how AI-supported content might influence reactions towards human communication, particularly in situations where it is made explicit that a human responder relied on AI to craft a response. However, this particular line of inquiry falls outside the scope of our current study, which aimed to evaluate AI and human responses independently, without external influences shaping the content produced by either agent. We wanted to preserve the distinctiveness of each agent’s unaltered capabilities and avoid imposing constraints on what or how much they could communicate. Introducing a human-AI collaborative element would have required additional control over the interaction that could potentially dilute the focus of this study—assessing the standalone qualities of AI and human responses.

That said, we completely agree that this is a fascinating and important area of research, particularly in contexts where AI-human collaborations could enhance empathy and response quality. In fact, we cite recent research by Sharma et al.

(2023) in our manuscript, where researchers specifically examine how AI-supported writing can improve the perceived empathy and effectiveness of human responses, particularly in support worker roles. While this intriguing direction goes beyond the immediate goals of our study, we believe it provides a valuable avenue for future exploration of human-AI collaboration in an empathic context even further.

We also apologize for any confusion regarding the presentation of our methods section. We ensured that the final version of the manuscript was clearly structured for ease of understanding.

11. "In sum, I think this is a super interesting question and a fascinating set of findings. However, I think you want to make sure you're not jumping the shark to ensure you are being true to what you can really conclude here, and so this work gets an appropriate and productive reception both academically and in the popular media."

We warmly thank the reviewer for their constructive critique of our work, which we believe became stronger following the implementation of their comments. We have reframed all sections of the manuscript to interpret the perceived effectiveness of AI in brief, written communicative contexts, at least by third party raters. We feel that these changes better reflect the results of our experiments, and we are confident in appropriately relaying the interpretation of these findings to interested readers.

Reviewer 3 Comments (Andrés Gvirtz)

12. "There is a lot to like in "The Kindness Machine: AI Outperforms Expert Humans in Expressing Compassion". The research contributes to the debate, by comparing how third parties judge AI and expert generated responses in regard to their compassion."

We are thankful to the reviewer for their expressed liking of our manuscript and for their acknowledgment of the manuscript's contribution to the debate surrounding AI-Human interactions, particularly within the context of demonstrated empathy.

13. "There is a lot of value in comparing the responses with experts, rather than layman, as in the context chosen, AI is suggested as a supplement or even as an alternative to experts, rather than of friendly layman advice. I have some questions regarding the overall framing, which I believe might profit from some tweaking, and a very basic additional analyses to suggest – all suggestions which I believe can be well addressed in a revision."

We are pleased to hear that the reviewer recognizes the value of comparing AI responses with experts' rather than non-expert responses to better understand

the utility of AI as a tool to enhance human communication in contexts where written support is needed. We have further revised the framing of the manuscript and ran a supplementary analysis to reflect the reviewer's following comments.

14. "A second named contribution is the source disclosure of the AI. I think this area is actually very well studied, especially in adjunct fields, such as communications and marketing, as well as computer science proceedings. I myself reviewed three different papers on similar topics, with every single one of them manipulating source disclosure. I believe this is important to do, but framing wise would focus a bit more on the expert element as a main contribution (for source disclosure you could relate it more to how it fits the existing literature, e.g. compared with much other work you seem to find less of a neg. effect after disclosure)."

We have cited the relevant literature to support our findings with respect to effects of source disclosure on AI responses' perceived compassion between **pages 17 and 18** in the discussion section, to clarify that this pattern of findings aligns with existing literature surrounding the impact of source disclosure on people's perceptions of AI-generated content. We write, "*This pattern of findings is consistent with the existing literature surrounding the impact of source disclosure on people's perceptions of AI-generated content, where the effectiveness of AI-generated content is lower when source identity is disclosed, even when its quality is evaluated as largely comparable to human responses*^{20,38,39}." (p. 18).

We further drew on more literature to emphasize our most notable finding, which is that AI responses outperformed experts' in terms of perceived compassion and were preferred to expert responses, on **pages 18 and 19** in the discussion. Specifically, we acknowledged the various constraints faced by our sample of crisis responders (as well as individuals in related professions or roles), which may consequently limit the quality of their empathic communication. We further drew on a recent study by Sharma et al. (2023), which explored the rated empathy of collaboratively written responses by support workers and AI on **page 19** to support AI's utility in complementing or enhancing human communication in brief, written contexts:

"In sum, despite the high quality of empathic responses from both samples of human responders, AI's consistent performance in providing superior empathetic responses highlights its utility in compassionate care through complementing, or potentially enhancing human communication⁸ and preparedness, particularly in brief, written contexts. This utility is underscored by recent findings demonstrating that support workers can successfully use AI collaboratively to guide their empathic responses⁴⁶."

15. “The focus of the paper is on empathy, with AI outperforming humans - a quick reader might take away that we can just rely on the AI in clinical/therapy contexts now. I think it is important to make explicit, that the main goal of a medical professionals is not always just being empathetic, and sometimes is not even the main goal, compared with e.g. treating the patient in an emergency situation. I am not at all suggesting empathy is not crucial, but I think one just needs to be careful in overall framing, as showing equivalence in empathy is a first, but very early step, with this research not aiming to evaluate if the given responses would actually lead to any desirable behaviour changes/outcomes etc. You mention that the LLM normally generate longer responses, and give as a potential reason the taxing nature of humans expressing empathy. This seems to be an empirically unsupported claim, and one that I would not expect to be the primary driver myself, having exploratively interviewed medical students and professionals in the past: short responses are probably much more driven by time constraints, and general demand, rather than by a lack of empathy from the provider, or an exhaustion of their empathy as they go through their day. Similarly, it was suggested that shortage of mental health workers is actually linked to fatigue of empathy, while this could be a partial reason, it seems a bit too strong of a claim to me, as it is currently presented as the main driver.”

We thank the reviewer for this very important comment. We have revised the way we refer to the empathetic offer by healthcare professionals to better reflect the complex reality that healthcare professionals are in. For example, on **page 3** in the introductory paragraph of the manuscript, we now write that, “*This seems to be particularly apparent in clinical settings, where healthcare professionals may sacrifice some of their ability to empathize in order to avoid burnout^{4,8}, to manage personal distress^{10,11}, or to balance their emotional engagement with the need to effectively allocate resources to each client. particularly individuals with complex cases¹².*”

We further clarify that: “*Such shortages make the maintenance of compassionate care even more difficult for the currently employed mental health professionals, for whom it serves as one of several key responsibilities⁸.*” (**p. 3**).

Additionally, we removed our rationale that the LLMs generating longer responses than physicians, as found in the Ayers et al. (2023), was a potential reason for the taxing nature of humans expressing empathy. Instead, on **page 4** in the introduction, we now write, “*Interestingly, chatbot responses were also significantly longer than physician responses, perhaps reflecting the difficulties for humans to convey empathy through written text, particularly when these responses are made by healthcare professionals, who may experience competing demands and time constraints^{12,21–23}.*”

In our evaluation of the differences in compassion ratings for AI and expert (crisis line worker) generated responses on **page 19**, we took care to note that, while

crisis responders receive quality training and may concurrently hold professional roles in various healthcare fields, *“These findings are particularly significant since responders in this organization undergo extensive training before selection⁴⁵ and may work concurrently in health fields that tend to centralize empathic communication with clients^{12,22,46}. Despite their training, these individuals report constraints like time pressures, high-severity cases²², and competing demands, which can contribute to compassion fatigue, staff shortages, and diminished client care and trust^{22,23}.”*

16. “Another framing related point, that I think you briefly touch on, the interactions are very short term. I believe this does not make them comparable to real therapy, but maybe more of a brief comforting situation. Therapy involves long term, contentious interactions. We are seeing AI’s memory increasing at the moment – so I am not suggesting that similar results shouldn’t be expected in therapy, but the study context for me is closer to the crises phone worker, than e.g. typical therapy. There is one sentence along these lines in the discussion, but I think being more explicit, and spending some more depth on here would be important for readers to be able to contextualise the work appropriately, as well as giving you the chance to guide future research.”

We thank the reviewer for this important point. We ensured that we removed any mentions of AI’s utility in therapy specifically within the manuscript, as our findings were neither reflected direct recipient feedback nor examined long-term written interactions between participants and responders This revision included the replacement of the ending sentence of the abstract to,

“These findings suggest that AI has robust utility in contexts requiring empathetic interaction, with the potential to address the increasing need for empathy in supportive communication contexts.” (p. 2).

Furthermore, we added the words “in brief written contexts” when interpreting AI’s utility with respect to expressions of empathy in the discussion. For example, we summarize a paragraph on **page 19** in the discussion section as such:

“In sum, despite the high quality of empathic responses from both samples of human responders, AI’s consistent performance in providing superior empathetic responses highlights its utility in compassionate care through complementing, or potentially enhancing human communication⁸ and preparedness, particularly in brief, written contexts. This utility is underscored by recent findings demonstrating that support workers can successfully use AI collaboratively to guide their empathic responses⁴⁶.”

17. “One methodological concern, that I don’t think could be solved easily: Your Study 1 and 2 samples are the result of picking the most empathetic out of a group, in Study 3 you use experts, i.e. crises workers, assuming their responses will be better. This

assumption seems to be unfounded to me, while I expect a randomly chosen participant to not outperform an expert, I actually expect the best out of a decently seized sample to outperform an expert. As we don't evaluate just outperforming, but rather outperforming in empathy specifically, the prediction gets more muddy to me, as an expert might follow other goals, too."

We thank the reviewer for this important point and agree that there are various factors that could contribute to the quality of empathic responses that professional empathizers such as crisis line workers may provide. To address this point, we responded to your suggestion by conducting an additional study, using the structure of study, to examine how third-party raters evaluated and expressed preferences for responses from expert and select responders to a sample of our empathy prompts (the detailed report of these findings can be found in the Supplementary Note 2 section of the Supplementary Information file). As the reviewer predicted, we found no significant difference in compassion ratings for expert and selected non-expert responders, and no preference for one or the other. We refer to these findings on **page 19** in the discussion to infer the constraints (e.g., time, competing demands) surrounding the provision of written empathic support by individuals working on a crisis line or related professions, and cited a recent study by Sharma et al. (2023) to tentatively support the utility of AI in collaborating with human support workers to enhance written communication and preparedness, given our findings from study 3. In short, our selected samples were as good at delivering empathic responses as our crisis line workers, which is consistent the reasoning of the reviewer—that the provision of empathy is one of numerous tasks crisis line workers need to accomplish, that time and other constraints can undermine this provision, and that a sample selected for the quality of their empathic responses is as good as (and could possibly be superior to) crisis line workers.

18. "Related, I would appreciate some more details for the methods, e.g. did the same participants do the prompts and the selection? How was it chosen which ones are most empathetic? What was the sample size of prompts, from which the best 10 were chosen etc.. Were groups instructed to be empathetic?"

We have updated our methods to specify the selection of the prompts. In the first paragraph of the methods section, we now mention that we (researchers) created the empathy prompts. On **page 7**, we now mentioned that an online sample of Prolific Academic workers created the responses used in studies 1 and 2. On **page 8**, we further clarify how the crisis responder (expert) and ChatGPT (AI) responses were generated and selected.

From paragraph 1 under methods: "*We first created 10 empathy prompts (first-person descriptions of both positive (5) and negative (5) experiences). In studies*

1-3, participants read all 10 empathy prompts describing personal experiences. In study 4, only 6 of the 10 prompts were presented to participants.” (p. 7).

From paragraph 2 under methods: “To generate the select human responses used in studies 1 and 2, we first piloted a study on our university participant pool and then formally ran the study on Prolific Academic³¹. Ten participants were instructed to read the 10 empathy prompts and generate a compassionate written response to the author of the prompt. Out of the 100 overall responses generated (10 per participant), we asked 3 graduate students and 4 research assistants to rank order the top 5 responders based on how overall compassionate their responses were in terms of quality, emotional salience, reliability, and level of detail. The 5 responders who were ranked in the top 5 most often had their responses selected for use in the studies. Thus, we consider these a select group of empathic responders, as they were first screened and selected based on their overall empathic quality.” (p. 7-8).

From paragraph 3 under methods: “In studies 3 and 4, the human response stimuli were obtained from a sample of hotline crisis responders—volunteers trained to respond to psychological crises through telephone calls—whom we considered expert empathizers. These participants were recruited via emails that were internally distributed to all responders within the Distress Centres of Greater Toronto, a multi-helpline organization that offers emotional support to Canadian callers across general and national helplines. Five responders provided written empathic responses to the same 10 empathy prompts as the Prolific Academic sample. All responses were inspected for quality and used in the study, such that each participant only saw one randomly selected option of these 5 responses per vignette.” (p. 8).

From paragraph 4 under methods: “The AI-generated responses across all studies were created by prompting ChatGPT (model gpt-4-0125-preview) with the 10 vignettes describing the emotional experience (one at a time) and asking it to generate an appropriate empathetic response. Given the stochastic nature of ChatGPT, we generated 5 separate responses per vignette. All responses generated by ChatGPT were used in the study. These responses were randomized in the study, such that participants only saw one of these 5 responses per vignette.” (p. 8).

19. “One additional analysis I believe could be useful in this context, is to have some between study comparisons. If I understand the set-up correctly, you have the same statements being judged partially in the skilled layman and in the expert condition, which means you could check the underlying assumption that experts are actually sig. better than selected layman at empathy.”

We thank the reviewer again for expanding on the way in which we could resolve the methodological concern surrounding the assumption that the expert responses in study 3 would be rated as significantly more compassionate than responses made by non-expert yet skilled responders. As we described above, we ran an additional study and reported it in our supplementary exploration of this assumption, where we found no significant differences in compassion ratings or preferences for expert and non-expert responses. Please see our response to **point 17** above for a detailed description of this assessment.

20. “On a more minor note, I found the entire article very pleasant to read and understandable, with the only exception being the abstract. Not having read the paper, I was confused in what would actually differentiate the conditions. Condition 1 is presented as skilled human responders, which I took to be as experts, but Condition 3 turned out to be the experts. Study 4 was then introduced as “transparency was kept constant”, which I found also a bit confusing at first. (Again it all made sense once I had read the introduction and paper, but was just a bit more difficult when I first read the abstract). Maybe some work could be undertaken here to make a reader, who is not familiar with the context or condition, aware of what each study is adding at-large.”

We thank the reviewer for the comment. We have updated the abstract to avoid confusion. We removed any mention of skilled human responders in the description of study 1, which was indeed misleading. We have also edited the entire methods section to ensure clarity about the differences in design and stimuli used across the studies.

Updated abstract: “[...] Results revealed that AI responses were preferred and rated as more compassionate compared to select human responders (Study 1). This pattern of results remained when author identity was made transparent (Study 2), when AI was compared to expert crisis responders (Study 3), and when author identity was disclosed to all participants (Study 4). Third parties perceived AI as being more responsive—conveying understanding, validation, and care—which partially explained AI’s higher compassion ratings in Study 4. These findings suggest that AI has robust utility in contexts requiring empathetic interaction, with the potential to address the increasing need for empathy in supportive communication contexts.

21. “On a very minor note, you mention that the average effect size in social psychology is .4? I remember more recent work, e.g. by Funder having found less than half, but might be mistaken.”

We thank the reviewer for this remark. We would like to clarify that we considered the average effect size based on Richard et al. (2003), who suggest that the average effect size is $r = .2$. Translated into Cohen’s d , that would be .4. While more contemporary analyses have returned smaller effects, this was what we based our power analysis on.

22. "Overall, I believe that the paper is interesting and is in a highly relevant context, and I am looking to read the revised version, for which I hope these comments are helpful."

We thank the reviewer for their kind words, their interest in our manuscript, and for their recognition of the relevance of our manuscript topic. We further express appreciation for their clear and thorough feedback, which have contributed to a strong improvement of our manuscript.

We hope that the revisions we have made adequately address your feedback and enhance the clarity, rigor, and overall contribution of our research. Once again, we thank you for your expertise and guidance throughout this process, and we look forward to your continued feedback.

Sincerely,

Dariya Ovsyannikova (on behalf of Victoria Oldemburgo de Mello and Michael Inzlicht)

Department of Psychology, University of Toronto

December 12, 2024

Dear Reviewers

We would like to once again express our sincere gratitude for your thorough and insightful feedback on our manuscript, now titled "Third-Party Evaluators Perceive AI as More Compassionate than Expert Humans" (Manuscript ID COMMSPSYCHOL-24-0351A), accepted in principle at *Nature Communications Psychology*. My co-authors and I greatly appreciate your thoughtful comments and suggestions, which were invaluable in guiding us to improve the manuscript.

Below, I present a summary of the point-by-point responses to the final feedback made by each reviewer, interweaving all review points with our responses.

Reviewer 1 Comments

1. I am satisfied with the authors' response and am looking forward to seeing this paper published.

We thank the reviewer for their thoughtful feedback. We are grateful for the time and effort they dedicated to reviewing our work and are pleased that our revisions addressed their concerns. Their positive assessment is deeply appreciated, and we are equally excited about the prospect of sharing this research with the journal readership.

Reviewer 2 Comments

2. I thank the authors for their conscientious response to my comments in the last round. All of my concerns have been addressed.

We thank the reviewer for their acknowledgment of our efforts to address their comments. We appreciate their constructive feedback throughout the review process, which has undoubtedly strengthened our manuscript. We are pleased that our revisions have resolved their concerns, and we value their thoughtful engagement with our work.

Reviewer 3 Comments (Andrés Gvirtz)

3. I thank the authors for the thorough, collaborative manner in which they addressed the previous round's comments. The revisions demonstrate a strong engagement with the feedback, particularly when it came to refining the framing, contextualizing the study slightly better within what is already known, and clarifying the methodological details. I also appreciate the additional study and analyses.

We thank the reviewer, Andrés Gvirtz, for their generous and encouraging feedback. We are grateful for their thoughtful and thorough suggestions, which guided us in refining the framing, contextualizing the study, and clarifying the methodology. We also appreciate their recognition of the additional study and analyses, which we believe have strengthened the manuscript. Their guidance has been invaluable, and we are pleased that the revisions meet their expectations.

4. The contribution is now slightly narrower, but I believe it is a better representation of the findings and implications, which I believe both to be meaningful. The 10 comments I previously had have all been addressed in a comprehensive manner, with the 7-page response to them being indicative of the work that went into addressing them. Overall, an interesting study that I am confident will add to the literature!

We agree that narrowing the contribution has allowed for a clearer and more accurate representation of the findings and their implications, and we are pleased that the revisions have improved the clarity and focus of our work. We appreciate the reviewer's acknowledgement of our efforts and are encouraged by their confidence in the study's potential to add to the literature.

Once again, we thank you for your expertise and guidance throughout this process, and we look forward to the formal acceptance of our manuscript, which would not have been possible without your efforts.

Sincerely,

Dariya Ovsyannikova (on behalf of Victoria Oldemburgo de Mello and Michael Inzlicht)

Department of Psychology, University of Toronto